# Enjoyment or Indulgence? Social Media Service Usage, Social Gratification, Self-Control Failure and Emotional Health

**DOI:** 10.3390/ijerph20021002

**Published:** 2023-01-05

**Authors:** Yan Liu, Hongfa Yi, Crystal Jiang

**Affiliations:** 1School of Journalism & Communication, Shanghai University, 149 Yanchang Drive, Shanghai 200072, China; 2Department of Media and Communication, City University of Hong Kong, Tat Chee Avenue, Hong Kong SAR, China

**Keywords:** mobile-based social media sites, social gratification, social media self-control failure, emotional health, app-tracking log data

## Abstract

Social networking site smartphone applications have been widely used among Chinese young adults. However, less is known about their effects on emotional health and the mechanisms through which they function. This study analyzes the relationship between college students’ smartphone social networking service use patterns, social gratification, social media self-control failure, and emotional health. Data was collected from 360 college students in China via application log tracking and a self-administered questionnaire. Structural equation modeling results showed that, after controlling for demographic variables, the use of video social networking site smartphone applications was associated with decreased social gratification, and ultimately, adverse emotional health. Using social networking site smartphone applications late at night exhibited worse emotional health via more social media self-control failure. The implications for designing and using social media applications are discussed.

## 1. Introduction

Technological advances have made social-networking services (SNS) easily accessible on smartphones, resulting in the widespread adoption of smartphone SNS applications, or ‘apps.’ SNS apps are top-rated among young users. In China, approximately 340 million individuals below the age of 30 use WeChat, while around 145 million users of Weibo are below 30 years of age [1]. Most SNS users use these networks on their smartphones [1]. These apps enable users to establish and maintain their social networks, obtain information, and seek entertainment like their web-based interfaces [2]. Given the large number of SNS apps use among Chinese young adults, it is critical to examine how SNS apps may affect their emotional health.

Studies have shown that using SNS brings benefits, including providing information, social support and entertainment [3,4]. Other studies have indicated that smartphone SNS use poses risks to users’ emotional health, including lower life satisfaction, depression, and anxiety [5,6,7,8]. The mixed results may be due to data inaccuracy [9]. Inaccuracy of SNS use does matter in understanding the effects on emotional health [10]. While prior studies have been based on self-reported data or observations of specific social media platforms (e.g., use frequency and duration) [7], few studies have thoroughly examined how SNS uses affect emotional health using behavioral data.

Scholars suggest that social gratification and social media self-control failure (SMSCF) are two important mechanisms through which SNS use affects emotional health. In particular, users may gain social gratification by interacting with others via SNS [11,12]. They may also experience negative emotions due to their internal failure on social media use self-control [13]. However, prior studies either focused on social gratification or SMSCF, neglecting that these two routes may occur simultaneously. The relative contributions of SNS use via social gratification or SMSCF to emotional health are not thoroughly explored.

Prior studies often treat SNS use as a homogeneous behavior, which may also contribute to the mixed findings. When exploring the effect of SNS app use, researchers often treat SNS app use as a general category [14,15], without considering specific conditions in which SNS apps are used (i.e., SNS app types, use time). SNS apps differ in types as they are conceptually designed differently, primarily serving different purposes. Thus, using different types of SNS may have diverse relationships with emotional health. Moreover, the time of using SNS apps may also impact emotional health differently [16]. These contextual factors (i.e., SNS app types and use time) should be considered when studying the effects of SNS app use on emotional health. 

It is noteworthy that prior studies on SNS app use and emotional health often relies on self-reported data. Such type of data may be inaccurate due to recall bias and social desirability [17]. Prior studies compared the self-reported SNS app use and their actual use, suggesting users tended to overestimate their SNS usage [17]. Moreover, such bias in SNS use is not random. It appears systematically related to users characteristics and has impacts on our assessment of the relationships between digital technology and emotional health outcomes [18]. For instance, recent studies found correlations between self-reported SNS app use and loneliness, life satisfaction, and depression were stronger than the correlations between actual behavioral records of SNS app use and these health outcomes [18]. In a related vein, a growing number of studies have employed smartphone log tracker apps to capture real-time behavioral data. Such tracker apps can accurately record the start time and end time whenever users use apps on their smartphones. Though such a big data approach provides detailed, enriched profiles of SNS app use data, it is often criticized for its atheoretical orientation and arbitrary variable selection [19]. Scholars thus call for combining self-reported and behavioral data to achieve more robust methodologies. 

In the present study, we employed a theoretical model, the inputs-mechanism-outputs model (I-M-O), to explore the relationship between different SNS use patterns, two important mechanisms (i.e., social gratification, SMSCF) and emotional health. We propose an analytical model that combines both behavioral data and self-reported data. We first use a computational method coding behavioral log data to accurately capture smartphone SNS use patterns. We also employ survey data to understand smartphone SNS user psychological factors (i.e., social gratification, SMSCF, and emotional health). This unique approach will deepen our understanding of the relationship between SNS use and emotional health, guide users to maximize their SNS use benefits, and bear implications for improving smartphone SNS designs.

## 2. Literature Review

### 2.1. The Inputs-Mechanism-Outputs Model

Chib and Lin propose the inputs-mechanism-outputs (I-M-O) model for understanding the effects of mobile health app use [20,21]. The inputs refer to mobile health app use. The mechanism draws on psychological, social, and behavioral theories to illuminate how the inputs may operate in health apps to produce adoption and adherence [22]. Outputs are concerned with users’ health outcomes resulting from the continuous use of mobile apps [20]. The merits of this model are that it incorporates technical and individual determinants specific to the mHealth context and flexibly integrates different theoretical lenses for synergic understanding. In the present study, we utilize the I-M-O model to explore how smartphone SNS use (i.e., input) is associated with emotional well-being (i.e., output) via social gratification and SMSCF (i.e., mechanism) (Figure 1).

As the primary gratification provided by SNS, social gratification brings immediate rewards through continuous, intensive, and satisfactory bonding [11,23,24]. Accumulatively, such social gratification is beneficial to emotional health. Studies have shown that online social connections can buffer anxiety and isolation [12]. However, these studies did not directly test the correlation between social gratification and emotional health. 

However, studies have shown that SMSCF may offset the immediate gratifications from SNS use, thereby being detrimental to emotional health [25]. SNS users may repeat failures in exercising self-control when using SNS [26]. Users may frequently face the strong temptation of SNS use even as it interferes with other important goals [27]. Users who may be driven primarily by immediate gratification and fail to consider the potential negative consequences of their actions experience SMSCF. Studies have shown that SMSCF is positively associated with procrastination, and ultimately associated with users’ academic stress and temporary mood swing [28], SNS addiction [29,30], and increased social anxiety and depression [29]. SMSCF disrupts users’ ability to achieve everyday personal goals such as quality sleep, study, or work performance [31], and compounds anxiety and stress through procrastination despite the urgency of tackling those important goals and tasks [28,29,30,31,32]. 

Though SNS users may simultaneously benefit from social gratification brought by personal interaction while impaired by their internal stress and anxiety brought on by SMSCF, studies have often tested the two mechanisms in separate models [e.g., 2, 26]). As external interactions and internal regulation are intertwined when using SNS, there is a critical gap in the research on SNS use and emotional health. In the present study, we examine the roles of social gratification and SMSCF in an integrated model that helps to explain the psychological consequences. Based on the prior literature, we propose:

**H1:** 
*Social gratification is positively associated with emotional health.*


**H2:** 
*SMSCF is negatively associated with emotional health.*


### 2.2. SNS Usage Patterns and Social Gratification

While SNS can bring social gratification, users may experience different levels of social gratification while using SNS, and different types of SNS offer users opportunities for social interaction at various levels. For instance, friend-networking SNS (e.g., Facebook) offer users a set of features (e.g., text, audio, pictures, video, and audio calls) to easily send and receive messages with others. Prior studies have consistently suggested that social gratification is the most important gratification for friend-networking SNS users [33]. By contrast, although social interaction functions (e.g., like, comment, and share) are also available in video-sharing SNS, such as Tiktok, users are less motivated to use these functions while immersed in video streaming [34,35,36,37]. Studies have shown that video SNS users tend only to use social interaction features to give comments, rather than interacting with other viewers [38]. Prior studies on Tiktok have shown that gratification of entertainment and affective needs, and the need for escapism, are important for using the platform [39,40,41]. Therefore, we propose:

**H2a:** 
*The use of friend-networking SNS (duration, frequency) is positively associated with social gratification.*


**H2b:** 
*The use of video SNS (duration, frequency) is negatively associated with social gratification.*


Information SNS are primarily designed for sharing and seeking information, but also with features that allow users to interact with each other. Prior studies found that information gratification was important for users of information SNS, whereas user social gratification seems to be less important [33]. Such information gratification is from the satisfaction of the quality of the information provided by SNS and the gratification of users sharing news with others. However, other studies show information SNS apps can still slightly meet users’ social gratification [33,42]. Therefore, we ask:

RQ1a: Are the uses of information (duration, frequency) associated with social gratification? 

A limited number of studies have explored the relationships between online discussion SNS and social interaction gratification. Prior studies on online discussion forums suggested that participants sought social support from others who had a similar experience, and shared their own thoughts and experience [43]. There should be social interactions among online discussion SNS participants by posting, replying, commenting, liking, and following. However, a number of studies also show participants in online discussion forums are to seek and share information, rather than interacting with group members [44]. Therefore, similar to information SNS, information gratification may be more satisfied than social gratification. We ask: 

RQ1b: Are the uses of online discussion SNS (duration, frequency) associated with social gratification? 

Moreover, levels of social interaction when using SNS may vary over a day. A person’s day and night are divided into periods, including paid work, housework, free time, and rest [45], and people often socially interact less during rest time. Therefore, we propose:

**H2c:** 
*The use of SNS during midnight (0–6 a.m.) is negatively associated with social gratification.*


### 2.3. SNS Usage Patterns and Social Media Self-Control Failure

While numerous studies have shown that SMSCF plays an important role in users’ emotional health [30], it remains unclear what specific patterns of SNS use affect users’ SMSCF. SNS use is a highly pleasurable activity that people are attracted to by its constant availability, persistent cues, and immediacy [27]. Insufficient self-control in SNS use is closely associated with Facebook addiction [46,47,48] and smartphone addiction [16,49]. One study found that low self-control users tend to be attracted to friend-networking SNS, but not SNS with an entertainment and information sharing focus [27,50].

**H3a:** 
*The uses of SNS (social-networking, online discussion, video, and information) are positively associated with SMSCF.*


Some studies have shown that using SNS very late at night is positively associated with sleep depletion [50,51], indicating a failure of self-control in using SNS during rest time. However, it is unclear how SNS usage during other time periods is associated with SMSCF. Therefore, we propose and ask:

**H3b:** 
*The use of SNS during midnight (0–6 a.m.) is positively associated with SMSCF.*


## 3. Methods

### 3.1. Participants

This study was approved by the Institutional Review Board of a Chinese university. We calculated the sample size with the following formula:$$n=\frac{{\left(z_{\frac{\alpha }{2}}\right)}^{2}\pi \left(1-\pi \right)}{E^{2}}$$

E is the sampling error. π is the proportion of the population, usually taken as 0.5 when $\pi \left(1-\pi \right)$ is maximized. The maximum sampling error allowed in social science is generally ±10%. We took the confidence level (1−α) as 0.95, and therefore the required sample size n≈100. This study focused on examining the underlying mechanisms by which SNS use may associate with emotional health. Therefore, convenience sampling was appropriate for this study. We had recruitment advertisements in several online discussion forums that were popular among several colleges and universities across mainland China. Students who were interested in this project would contact our research assistants. After participants signed the consent forms, they were asked to download the smartphone tracker app. The inclusion criteria were: (1) College students and (2) Android smartphone users. Participants were asked to download an app named ‘App Usage Tracker’ on their smartphone and keep the app tracker running throughout the data collection period. Data from 372 college students (sampling error  ≈ ± 5.08%) across mainland China were collected between May 2019 and November 2020. Two graduates pre-tested the tracker app on the participant’s smartphone and assisted in transmitting the smartphone app tracker data after use for one week (Monday to Sunday). After sharing the smartphone app tracker data, the participants responded to an online survey. Log and survey data were merged in each participant’s unique identifier and recorded in all data files. Participants were compensated with the equivalent of USD$7.00 each upon completion of the study. Twelve participants were excluded due to incomplete app tracker data, leaving 360 participants included in the study. Of the participants, 46.4% were male, 85.8% were Han Chinese, and 48.1% were freshmen. Their monthly expenditure on food and non-essentials were USD$200 (RMB1, 290) (Table 1).

### 3.2. Social-Networking Application Log Data

The tracking app automatically recorded the app usage in the usage logs. The log data included: (a) The name of the app running in the foreground, (b) app start date and time, and (c) app use duration in seconds.

*App category coding*. The 360 participants used a total of identical 1532 apps. Two graduates reviewed app descriptions on Google Play Store using the app names, checked their functions and completed a training session with pilot data. The interrater reliability was high, with a Cohen’s Kappa of 0.96. First, apps were coded into social networking (i.e., SNS) apps, and non-social networking apps. SNS apps refer to apps that allow users to share ideas, digital photos and videos, posts, and to inform others about online or real-world activities and events (e.g., WeChat, Weibo, Hupu, and Tiktok). Non-SNS apps refer to apps that are not designed for social networking purposes (e.g., Adobe Photoshop, Word, and Map). Second, SNS apps were further coded into four categories according to their primarily-designed functions: Friend-networking, video, information, and online discussion. Friend-networking SNS refers to SNS apps that focus on bonding (e.g., QQ, WeChat); video SNS refers to SNS apps that were designed for short-form video sharing (e.g., Tiktok, Kuaishou); information SNS refers to SNS apps designed mainly for creating, distributing and discovering information (e.g., Weibo); and online discussion SNS refers to SNS apps that allow users to post and reply in online discussion forums (e.g., Hupu, Baidu Tieba). 

*Time spent*. The total hours spent on each SNS app were the mean of time spent on each type of SNS app, and the mean of time spent during each time period across a day for each user.

*Use frequency.* The number of times each type of SNS app was used and the number of times SNS apps were used during each time period across a day were calculated for each user.

### 3.3. Survey Data

*Emotional health.* The participants were asked to rate the extent to which they had experienced each mood state during the previous week, including anger, sadness, and anxiety states [52] on a 7-point scale (1 = Very slightly or not at all, a little; 7 = Very much). 

*Social gratification from SNS use.* A 4-item measurement tool developed by Whiting and Williams [53] was employed to measure the participant’s social gratification from using SNS (e.g., “I use social media to connect and stay in touch with my family and friends”). The participants responded using a 5-point Likert scale (1 = Strongly disagree; 5 = Strongly agree). 

*Social media self-control failure.* A 3-item social media self-control failure (SMSCF) scale [13] was used to measure the participant’s level of self-control when using SNS (e.g., “How often do you give in to the desire to use social media even though your social media use at that particular moment causes you to utilize your time less efficiently”). The participants responded using a 5-point Likert scale (1 = Almost never; 5 = Very often).

The Chinese version of social gratification and SMSCF were developed using the back-translation method. First, the original version was translated into Chinese by one bilingual translator and was back-translated into English by another bilingual translator. Finally, the original version of the social gratification and SMSCF were compared with the back translations. If discrepancies arose in the back-translation, translators worked cooperatively to make corrections to the Chinese version. 

The participant’s demographic information, including gender, race, college year, and monthly expenditure on food and non-essentials, was also recorded in the survey.

### 3.4. Statistical Analysis

A confirmatory factor analysis (CFA) was conducted to get the validity and reliability of three latent variables: Emotional health, social gratification, and SMSCF. To test our hypotheses and answer our research questions, a series of structural equation models were conducted with the Mplus 8.1 model (https://www.statmodel.com/, accessed on 15 November 2022). Exogenous variables were: (1) Use frequency, (2) time spent on the four types of SNS application (i.e., friend-networking, video, information, and online discussion), and (3) use frequency and time spent during four time periods during a day (i.e., 0–6 a.m., 6 a.m.–12 p.m., 12 p.m.–6 p.m., and 6 p.m.–12 a.m.). The endogenous variables were: (1) Social gratification, (2) SMSCF, and (3) emotional health. Social gratification and SMSCF were also mediators. The model fit was assessed using the chi-square (χ2), root mean square error of approximation (RMSEA), comparative fit index (CFI), standardized root mean square residual (SRMR), and TLI (Tucker-Lewis index). Gender, race, college year, and monthly expenditure on food and non-essentials were treated as control variables. 

## 4. Results

### 4.1. Descriptive Statistics

Table 1 illustrates the demographic details of the respondents. Table 2 shows SNS use frequency and duration time on four types (i.e., friend-networking, video, information, and online discussion) and during a day (i.e., 0–6 a.m., 6 a.m.–12 p.m., 12 p.m.–6 p.m., and 6 p.m.–12 a.m.).

### 4.2. Measurement Model

The internal reliability of the measurement model was found to be satisfactory (Table 3). The fit indices of the measurement model were satisfactory χ^2^/df = 3.255, CFI = 0.930, TLI = 0.902, SRMR = 0.053, RMSEA = 0.095 (Table 4). Common method bias (CMB) was added to the measurement model as a latent variable, and the measurement model with CMB has constructed accordingly. Compared with the measurement model, the fit indices of the measurement model with CMB were not better, which indicates there was no significant common method bias. The reliability of the study measures (social gratification, SMSCF, and emotional health) was examined using Cronbach’s α and the composite reliability (CR) coefficient. The factor loading was above 0.5. Values for Cronbach’s α exceeding 0.70, and CR values exceeding 0.60, were considered indicative of adequate internal consistency. In addition, we also looked at the average variance extracted (AVE) for each latent variable to ensure that items were contributing adequately to the construct they indicated, with values in excess of 0.50 considered satisfactory.

**Table 2 ijerph-20-01002-t002:** Description of SNS use duration and use frequency.

Variable	Mean	SD	Min	Max	N
**SNS category use**					
Time spent on friend networking SNS	2.879	2.814	0.142	17.566	360
Time spent on video SNS	0.673	1.400	0.000	9.225	360
Time spent on information SNS	0.454	0.996	0.000	9.420	360
Time spent on online discussion SNS	0.143	0.397	0.000	4.039	360
Frequency of friend networking SNS use	68.993	63.218	3.286	415.571	360
Frequency of video SNS use	4.191	8.793	0.000	88.571	360
Frequency of information SNS use	5.814	11.732	0.000	121.143	360
Frequency of online discussion SNS use	2.125	6.239	0.000	63.429	360
**SNS use across a day**					
Time spent on SNS from 0–6 a.m.	0.287	0.502	0.000	3.076	360
Time spent on SNS from 6–12 p.m.	1.033	0.993	0.000	5.145	360
Time spent on SNS from 12–18 p.m.	1.379	1.305	0.005	6.853	360
Time spent on SNS from 18–24 p.m.	1.475	1.377	0.018	6.974	360
Frequency of SNS use from 0–6 a.m.	3.198	6.465	0.000	86.714	360
Frequency of SNS use from 6–12 p.m.	20.206	19.255	0.143	125.714	360
Frequency of SNS use from 12–18 p.m.	27.867	24.917	0.429	168.000	360
Frequency of SNS use from 18–24 p.m.	30.750	28.102	0.857	154.286	360

### 4.3. Research Model and Hypothesis Tests

The structural models indicated that the fit indices of the research models were satisfactory: Model 1 assessed the relationships between time spent on different SNS app types and emotional health. χ^2^/df = 2.594, CFI = 0.904, TLI = 0.871, SRMR = 0.049, RMSEA = 0.067. Model 2 assessed the relationships between the use frequency of different types SNS apps and emotional health, χ^2^/df = 2.602, CFI = 0.903, TLI = 0.870, SRMR = 0.051, RMSEA = 0.067. Model 3 assessed if time spent on SNS app across a day associated with emotional health, χ^2^/df = 2.645, CFI = 0.901, TLI = 0.867, SRMR = 0.053, RMSEA = 0.068. Model 4 assessed the associations use frequency of SNS apps across a day and emotional health, χ^2^/df = 2.526, CFI = 0.907, TLI = 0.876, SRMR = 0.051, RMSEA = 0.068 (Table 4). The structural models are shown in Figure 2a–d.

### 4.4. Hypothesis Testing

Figure 2a showed the relationships among time spent on video SNS was negatively associated with social gratification (β = −0.130, *p* < 0.05), whereas social gratification was positively associated with emotional health (β = 0.158, *p* < 0.05); SMSCF was negatively associated with emotional health (β = −0.307, *p* < 0.001). Figure 2b. showed information SNS use frequency was positively associated with social gratification (β = −0.143, *p* < 0.05), whereas video SNS was negatively associated social gratification (β = −0.116, *p* < 0.05). Social gratification was positively associated with emotional health (β = 0.159, *p* < 0.05); SMSCF was negatively associated with emotional health (β = −0.307, *p* < 0.001). Figure 2c. showed time spent on SNS from 0–6 was positively associated with SMSCF (β = 0.144, *p* < 0.05). Social gratification was positively associated with emotional health (β = 0.158, *p* < 0.05); SMSCF was negatively associated with emotional health (β = −0.307, *p* < 0.001). Figure 2d showed SNS use frequency from 0–6 was positively associated with SMSCF (β = 0.170, *p* < 0.05). Social gratification was positively associated with emotional health (β = 0.157, *p* < 0.05); SMSCF was negatively associated with emotional health (β = −0.306, *p* < 0.001). 

For social gratification, SMSCF and emotional health, H1 and H2 were supported; social gratification led to better emotional health (β = 0.158, *p* < 0.05, Figure 2a; β = 0.159, *p* < 0.05, Figure 2b; β = 0.157, *p* < 0.05, Figure 2c,d), and SMSCF had a negative effect on subjective wellbeing (β = −0.307, *p* < 0.001, Figure 2a,b,d; β = −0.306, *p* < 0.001, Figure 2c).

For SNS use and social gratification, consistent with H2b, video SNS use was negatively associated with social gratification (β = −0.130, *p* < 0.05, Figure 2a; β = −0.116, *p* < 0.05, Figure 2b). H2a was rejected. For RQ1a, we found that information SNS use frequency was positively associated with social gratification (β = −0.143, *p* < 0.05, Figure 2b). For RQ1b, we did not find any association between online discussion SNS use and social gratification. H2c was rejected.

For SNS use and SMSCF, H3a was rejected. H3b was supported; using SNS during midnight (0–6 a.m.) was negatively associated with SMSCF (β = 0.144, *p* < 0.05, Figure 2c; β = 0.170, *p* < 0.05, Figure 2d). Table 5 summarized the results of the structural models.

## 5. Discussion

### 5.1. Summary of Major Findings

Combining log data and survey data, this study employed the I-M-O model to examine the relationship between SNS use patterns, social gratification, SMSCF, and emotional health. Study results showed that video SNS app use was negatively associated with social gratification and ultimately negatively associated with emotional well-being. Information SNS app use was positively associated with social gratification. Using more SNS during early morning hours (0–6 a.m.) was more likely to deteriorate emotional well-being via SMSCF.

We used an analytical model that follows the mixed-method approach. This approach integrated computational methods and one online survey, which can capture both individual actual behavioral data and the individual latent factors (i.e., social gratification, SMSCF, and emotional health). We also conducted the present study under the guidance of the theoretical framework (i.e., I-M-O), which helps researchers understand and interpret the relationships in depth. The present study provides an example for future studies aimed at understanding the impact of SNS use using big data and surveys together with the guidance of theory.

### 5.2. Theoretical Implications

The findings provide novel information regarding smartphone use in specific contexts (i.e., SNS use types, use times) and the relationships with emotional health. While existing literature paid much attention to the general SNS app use, our study goes beyond the existing literature by distinguishing different SNS app use by SNS app types and use time periods across a day. These differentiations help researchers to study the relationships between SNS app use and emotional health in a detailed way.

Different SNS use patterns were associated with distinct emotional health via social gratification. Using video SNS was negatively associated with emotional health via decreased social interaction gratification. Our findings suggest that using video SNS may either not satisfy users’ social interaction needs or present a passive and isolating experience [36,39]. For instance, although users can use many features to express their reaction to the video contents (e.g., like, share, and comment), they may not receive feedback from other users. The lack of reciprocity may relate to the unsatisfied social interaction need, further associated with decreased psychological well-being. Moreover, video watching can be used to escape real life [34]. While watching a video, users may transport into the narrative stories [37]. Either consciously or unconsciously, while using video SNS, users are not interacting socially with others as actively as they are when using other types of SNS. This unmet social gratification partially explained the negative association between excessive video SNS use and emotional well-being.

Furthermore, the study also found a link between different SNS usage patterns and SMSCF. Specifically, smartphone SNS usage during periods set aside for rest was associated with increased SMSCF, while SNS usage during other periods was not. These findings are consistent with related literature showing that smartphone SNS usage during late-night hours causes sleep depletion or sleep disturbance [14,15,16,17,18,19,20,21,22,23,24,25,26,27,28,29,30,31,32,33,34,35,36,37,38,39,40,41,42,43,44,45,46,47,48,49,50,51]. This result further provides empirical evidence that using SNS late at night can affect emotional health via increased SMSCF. Interestingly, such a detrimental link held for all types of SNS apps. As SNS sites have complete functions, SNS platforms may not vary significantly at providing constant availability, persistent cues, and other features that solicit SMSCF [27]. Taken together, regardless of the types of SNS, when SNS is used seems to be more crucial [7].

Finally, this study offers a comprehensive understanding of the health effects of SNS use by illuminating the mechanisms with the I-M-O model. Our study first provides a direct and simultaneous test of pathways from detailed smartphone SNS usage to emotional health impacts with two underlying and often conflicting mechanisms—social gratification and SMSCF. The results were consistent with prior studies showing users who get more social gratification [12] and less SMSCF are more likely to have better emotional well-being [29]. Going beyond the existing literature, this study use data obtained from daily actual SNS app use, which accounts for contextualized facts, showing that users get benefits from gratification but harms from concurrent internal self-control failure. The findings suggest, to maximize the long-term social gratification benefit of using SNS on emotional health, users need to balance between social gratification and internal self-control.

These findings have practical implications. First, campaigns can be implemented to improve digital literacy SNS usage habits among young adults. For instance, they should be instructed to monitor for excess uses during sleep time or replace SNS use with other calming activities before bedtime, such as taking a bath or using relaxation techniques, to avoid SMSCF. Second, users may consider adopting effective strategies to decrease their self-control failure with excessive SNS usage [54]. It is also worth noting that there may be a difference between SMSCF and personal trait self-control. An individual with strong self-control may not automatically be capable of controlling one’s social media use when it conflicts with other tasks and obligations (e.g., sleep). Intervening strategies can be used. For instance, smartphone notifications could be switched off before bedtime to reduce the likelihood of SNS checking [27]. Third, SNS designers may customize existing features and design new features according to usage patterns; for example, adding Chabot feedback for social gratification in video streaming.

### 5.3. Limitations and Future Directions

This study has certain limitations that could be addressed in future research. First, conclusions could not be drawn regarding the causal relationships among the measured variables. Users with emotional health problems may spend more time using SNS during regular rest times [55]. It is likely that users who have less social gratifications tend to use more video SNS apps [56]. It also may be a self-perpetuating process. For instance, those with lower emotional health tend to use SNS at late night and have even lower emotional health since they may feel guilt and anxiety that derived from SMSCF. Future studies can employ other research methods (e.g., longitudinal, and experiment) to explore the relationships. Second, our study sample was composed of university students using Android phones, with iPhone users excluded because tracking their usage data would have required downloading a separate app-specific for iPhone users. Therefore, it would be beneficial to replicate this study using other groups, including iPhone users. As Android phones take 91.2% of the smartphone share in China smartphone market [57], the SNS use patterns captured in this study have mostly reflected how smartphones are used in Chinese young adults.

Future studies could consider other patterns of SNS usage, such as quick-switching between different SNS apps, or remaining for long periods in one app—usage patterns that may also impact emotional health via social gratification and SMSCF. In this study, we distinguished different SNS types and time periods when examining the SNS app use phenomenon. With advance of behavioral data, researchers can further categorize SNS app use in other ways. For instance, before regular bedtime, how individual users switch between SNS apps (e.g., from information to friend-networking, from video to information), and how these switches impact their physical (e.g., sleeping quality) and emotional health.

Future studies could also consider tracking SNS use for a long study period and measuring SMSCF and social gratification multiple times to understand the long-term relationships between SNS use and emotional health. Though we collected the data over a one-week period, our findings uncovered the relationships among different SNS use patterns, two mechanisms and emotional welling being. It is possible that users switch their SNS use patterns or improve their SMSCF by employing intervening strategies over time. To understand the accumulative effect of SNS use and emotional health, longitudinal studies may be needed.

## 6. Conclusions

Very limited work has addressed the heterogeneity in SNS app use, and these use patterns associated with emotional health via external interaction (i.e., social gratification) and internal regulation (i.e., SMSCF). As such, this study provides an important step toward understanding these issues. The results clarify the conditions under which SNS use is associated with emotional health via social gratification and SMSCF. Insofar as smartphone SNS use grows at a staggering rate, these findings may contribute to a better understanding of how designers can monitor and structure interactions to minimize the drawback for smartphone SNS users.

## Figures and Tables

**Figure 1 ijerph-20-01002-f001:**
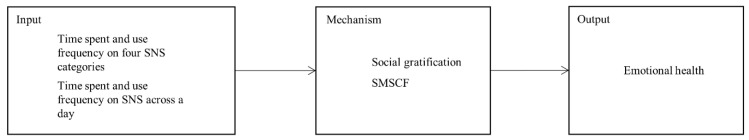
Conceptual framework of the present study.

**Figure 2 ijerph-20-01002-f002:**
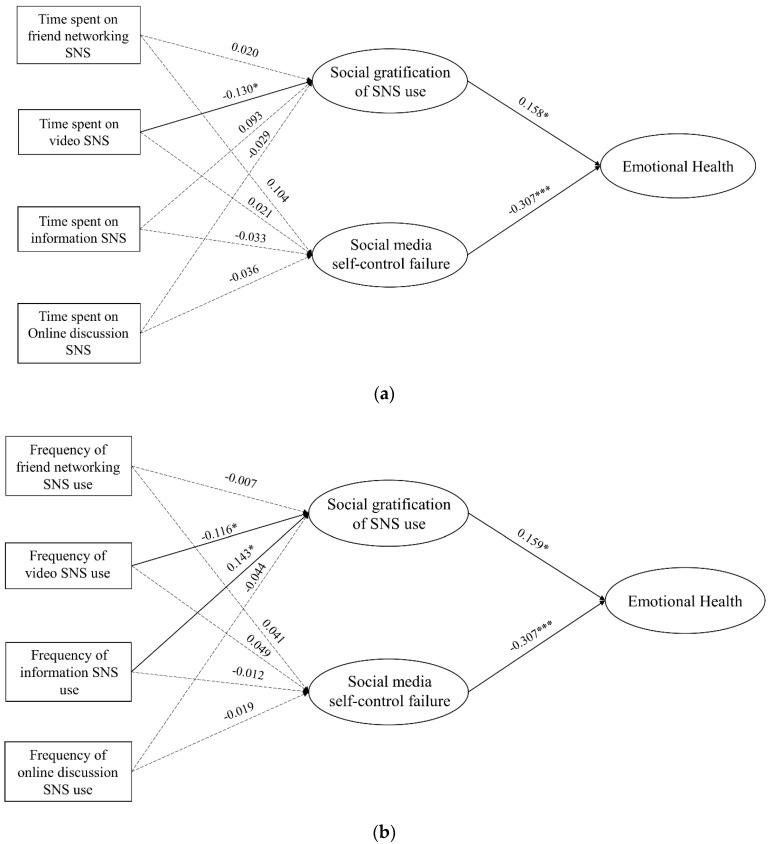
(**a**) Structural equation model 1 predicting emotional health from social gratification, social media self-control failure, and time spent on SNS with standardized path coefficients *(** *p* < 0.05, *** *p* < 0.001). (**b**) Structural equation model 2 predicting emotional health from social gratification, social media self-control failure, and SNS use frequency with standardized path coefficients (* *p* < 0.05, *** *p*< 0.001). (**c**) Structural equation model 3 predicting emotional health from social gratification, social media self-control failure, and time spent on SNS across a day with standardized path coefficients (* *p* < 0.05, ** *p* < 0.01, *** *p* < 0.001). (**d**) Structural equation model 4 predicting emotional health from social gratification, social media self-control failure, and SNS use frequency across a day with standardized path coefficients (* *p* < 0.05, ** *p* < 0.01, *** *p* < 0.001).

**Table 1 ijerph-20-01002-t001:** Participant demographics.

Characteristics	N (Percentage) or M (SD)
Gender	
Male	167 (46.4%)
Female	193 (53.6%)
Race	
Han	309 (85.8%)
Minority	51 (14.2%)
College year	
Freshman	173 (48.1%)
Non-freshman	187 (51.9%)
Monthly expenditure on food and non-essentials (unit: thousand RMB)	1.29 (0.58)

**Table 3 ijerph-20-01002-t003:** Results of the confirmatory factor analysis.

Construct	Item	Standardized Factor Loading	Cronbach’s α	CR	AVE
Emotional health	EH1	0.645	0.789	0.796	0.571
	EH2	0.689			
	EH3	0.906			
Social gratification of SNS use	SG1	0.548	0.821	0.826	0.556
	SG2	0.552			
	SG3	0.881			
	SG4	0.918			
Social media self-control failure	SF1	0.416	0.759	0.796	0.588
	SF2	0.917			
	SF3	0.866			

Note: CR = composite reliability; AVE = average variance extracted. EH1 = “I experienced anger during the previous week”; EH2 = “I experienced sadness during the previous week”; EH3 = “I experienced anxiety during the previous week”; SG1 = ”I use social media to contact people I don’t regularly see”; SG2 = “I use social media to make new friends”; SG3 = “I use social media to connect and stay in touch with my family and friends”; SG4 = “I use social media to stay connected with others”; SF1 = “How often do you give in to the desire to use social media even though your social media use at that particular moment conflicts with other goals”. SF2 = “How often do you give in to the desire to use social media even though your social media use at that particular moment makes you use your time less efficiently”; SF3 = “How often do you give in to the desire to use social media even though your social media use at that particular moment makes you delay other things you want or need to do”.

**Table 4 ijerph-20-01002-t004:** Fit indices of the measurement and structural models.

	Measurement Model	Measurement Model with CMB	Structural Models			
			Model 1	Model 2	Model 3	Model 4
**Chi-square/df**	3.255	3.536	2.594	2.602	2.645	2.526
**RMSEA**	0.095	0.099	0.067	0.067	0.068	0.065
**SRMR**	0.053	0.052	0.049	0.051	0.053	0.051
**CFI**	0.930	0.929	0.904	0.903	0.901	0.907
**TLI**	0.902	0.893	0.871	0.870	0.867	0.876

**Table 5 ijerph-20-01002-t005:** Results of the structural models.

Model	Path			Standardized Path Coefficients
Model 1	Time spent on video SNS	→	Social gratification of SNS use	−0.130 *
	Social gratification of SNS use	→	Emotional Health	0.158 *
	Social media self-control failure	→	Emotional Health	−0.307 ***
Model 2	Frequency of video SNS use	→	Social gratification of SNS use	−0.116 *
	Frequency of information SNS use	→	Social gratification of SNS use	0.143 *
	Social gratification of SNS use	→	Emotional Health	0.159 *
	Social media self-control failure	→	Emotional Health	−0.307 ***
Model 3	Time spent on SNS from 0–6 a.m.	→	Social media self-control failure	0.144 **
	Social gratification of SNS use	→	Emotional Health	0.157 *
	Social media self-control failure	→	Emotional Health	−0.306 ***
Model 4	Frequency of SNS from 0–6 a.m.	→	Social media self-control failure	0.170 **
	Social gratification of SNS use	→	Emotional Health	0.157 *
	Social media self-control failure	→	Emotional Health	−0.307 ***

Note: * *p* < 0.05, ** *p* < 0.01, *** *p* < 0.001. Insignificant results were removed.

## Data Availability

The data that support the findings of this study are available from the corresponding author, Hongfa Yi, upon reasonable request.

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
