# Peer review of "Enjoyment or Indulgence? Social Media Service Usage, Social Gratification, Self-Control Failure and Emotional Health"

_ijerph, 2023, doi:10.3390/ijerph20021002_

Round 1

Reviewer 1 Report

Inconsistency between Lines 202 and 272. Line 202 says "Table 1.a" and Line 272 says "Table 1.b."  I think "Table 1a." and "Table 1b." would be clearer, since without a period "a" looks like an article when none is needed, as in "a Participant Demographics" (Line 202). This is a minor detail. 

My major concern is that the results of Figures 2a-d are not immediately graspable, yet you make "unexplained" conclusions, such as Line 306 "H2 a was rejected." I assume you rejected it because -.007 value in Figure 2b, but I'm just "guessing" here. I would change Section 4.4, so that you begin by explaining what each of the 4 figures indicates in one sentence, before going into details. I see that you have a darkened line to indicate relevant findings. That is helpful for the figure, but the text needs clarification. Since the goal of the apps is to keep people attracted as long as possible, I don't see app designers using this publication to make changes to their products. 

It seems to me that low self-control would be a natural indicator for a) need for social gratification and b) low emotional health. I thus worry that all of this data merely deals with the symptoms and not the causes (low self-control/lack of discipline). Research ought to begin with well-being/emotional health, rather than end with it, since those who are unaffected are likely to already have higher emotional health and therefore the ability to be more disciplined, because they don't feel deprived and desperate.  

Reviewer 2 Report

Study Title: Enjoyment or Indulgence? Social media service usage, social gratification, self-control failure, and emotional health

The current study investigated how social media service usage might affect one’s emotional health through the social gratification and self-control failure pathways. The results from 360 college students show negative correlations between the use of video SNS and social gratification which ultimately, have adverse effects on emotional health. Moreover, using SNS late has negative effects on emotional health through social media self-control failure. The manuscript is well-written and insightful in general, though might benefit from some modifications.

Introduction

·         In the second paragraph, the author wrote “Though some studies have shown that using SNS brings benefits, other studies have indicated that smartphone SNS use poses risks to users’ emotional health”. Could the author be more specific about the benefits and risks for the users?

Literature Review

·         The authors did a great job reviewing the literature; however, the structure is a little confusing. Could the author make sure to link each research question to their hypotheses?

Methods

·         Is there a reason that the author used a 7-point scale for emotional health, a 4-point scale for social gratification, and a 3-point scale for SMSCF? Could the author explain more about how they make the scales comparable?

·         Did the authors consider putting the demographic information as covariates in the analyses? It would be interesting to see how demographic information might affect the associations.  

Results

·         In Table 2, could the author explain what the items stand for each construct (e.g., EH1, EH2, etc.)? Maybe the author could add this information as footnotes.

Discussion

·         The author investigated the short-term relationships between SNS usage and emotional health within the experimental period of a week. Could the author discuss how SNS usage may affect emotional health in a long term? Maybe the author could give recommendations for future studies that focus on the long-term effect of SNS usage.

Minor edits

·         Please explain the abbreviation (i.e, SNS) in the abstract.

·         Tables 1 a and b provided very different information, one for participant demographics and one for the descriptions of SNS use. Could the author name the table in a continuous manner instead of dividing them into a and b?

·         The arrangement of Table 1 b is off. I recommend only showing numbers with 2 decimal points so the numbers could stay in a single row.

Reviewer 3 Report

While the research is interesting, there are several aspects that can be improved. hence, here are some notes:

1. How is the sample size calculated?

2. What type of sampling methods has been used?

3. What was done in regard to CMB?

4. Hypothesis testing is shown by figures, however,  it would be more appropriate if the table shows.
